# Network controllability of structural connectomes in the neonatal brain

Huili Sun [1] ✉, Rongtao Jiang [2], Wei Dai[3], Alexander J. Dufford[4], Stephanie Noble [5,6,7], Marisa N. Spann[8,9], Shi Gu[10,11] & Dustin Scheinost [1,2,12,13,14] ✉

White matter connectivity supports diverse cognitive demands by efficiently constraining dynamic brain activity. This efficiency can be inferred from network controllability, which represents the ease with which the brain moves between distinct mental states based on white matter connectivity. However, it remains unclear how brain networks support diverse functions at birth, a time of rapid changes in connectivity. Here, we investigate the development of network controllability during the perinatal period and the effect of preterm birth in 521 neonates. We provide evidence that elements of controllability are exhibited in the infant's brain as early as the third trimester and develop rapidly across the perinatal period. Preterm birth disrupts the development of brain networks and altered the energy required to drive state transitions at different levels. In addition, controllability at birth is associated with cognitive ability at 18 months. Our results suggest network controllability develops rapidly during the perinatal period to support cognitive demands but could be altered by environmental impacts like preterm birth.

The structural connectome is a comprehensive map of the anatomical white matter connections studied as a complex network. It constrains brain dynamics in children, adolescents, and adults[1,2]. An infant's structural connectome develops rapidly during the perinatal period (approximately from the 28th week of gestation through the first month of postnatal life). This development establishes the foundation for later developing brain dynamics[3–5]. For example, a rich club of interconnected cortical hubs and other topological properties of the brain are already present by 30 weeks of gestation[6–8]. Similarly, infants can be identified from repeated scans using structural—but not functional—connectomes, suggesting that the structural connections

develop before associated function connections[9–11]. However, these measures lack a mechanistic description of how these structural changes support later brain dynamics. The question "How does an infant's structural connectome develop to constrain later brain dynamics?" remains unanswered.

Network control theory (NCT) provides a formal framework to study how brain dynamics arise from the structural connectome[12]. Controllability is an energy-based measurement, representing the ease of switching between different brain states[13]. This control energy is associated with gray matter integrity, glucose metabolism[14], and efficiency in cognitive execution[15,16]. In practice, it can be operationalized

[1]Department of Biomedical Engineering, Yale University, New Haven, CT 06520, USA. [2]Department of Radiology & Biomedical Imaging, Yale School of Medicine, New Haven, CT 06510, USA. [3]Department of Biostatistics, Yale School of Public Health, New Haven, CT 06510, USA. [4]Department of Psychiatry and Center for Mental Health Innovation, Oregon Health & Science University, Portland, OR 97239, USA. [5]Department of Psychology, Northeastern University, Boston, MA 02115, USA. [6]Department of Bioengineering, Northeastern University, Boston, MA 02115, USA. [7]Center for Cognitive and Brain Health, Northeastern University, Boston, USA. [8]Department of Psychiatry, Vagelos College of Physicians and Surgeons, Columbia University, New York, NY 10032, USA. [9]New York State Psychiatric Institute, New York, NY 10032, USA. [10]School of Computer Science and Engineering, University of Electronic Science and Technology of China, Chengdu, China. [11]Shenzhen Institute for Advanced Study, University of Electronic Science and Technology of China, Shenzhen, China. [12]Department of Statistics & Data Science, Yale University, New Haven, CT 06520, USA. [13]Child Study Center, Yale School of Medicine, New Haven, CT 06510, USA. [14]Wu Tsai Institute, Yale University, 100 College Street, New Haven, CT 06510, USA. ✉e-mail: huili.sun@yale.edu; dustin.scheinost@yale.edu

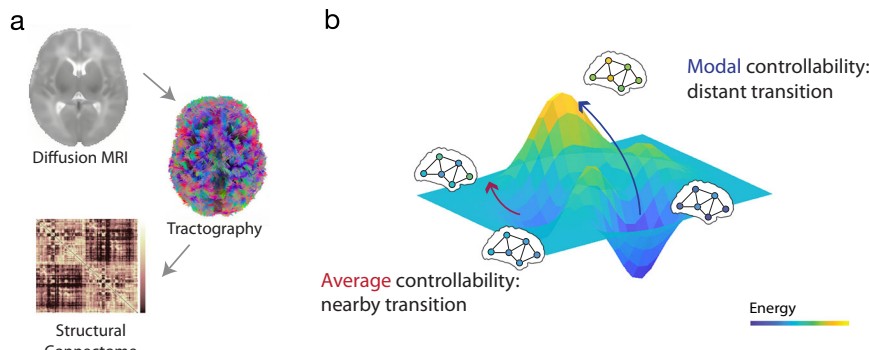

**Fig. 1 | Network control theory. a** Using diffusion-weighted imaging (DWI), structural connectomes were created from automatic fiber tracking for 521 infants. From these connectomes, average and modal controllability were calculated. **b** Controllability represents the ease of switching between different dynamic brain states. Average controllability measures a regional capability to support nearby state transitions. Modal controllability measures a regional capability to support distant state transitions.

as two dependent but complementary measurements. Average controllability measures the connectome's ability to drive the brain toward nearby brain states. Modal controllability measures the connectome's ability to move the brain toward distant brain states. Regions in the default mode network facilitate transitions to nearby states in adults (i.e., higher average controllability). Whereas areas in the frontoparietal network facilitate state transitions to far away states (i.e., higher modal controllability)[17]. NCT is also applicable to studying brain development. Controllability across the whole brain matures during adolescence, reducing the theoretical energetic costs of transitions to activation states necessary for adult-level executive functioning[15,18,19]. These results highly suggest that a more controllable structural connectome facilitates the brain dynamics needed for more cognitively demanding tasks.

We leveraged NCT to address three questions about how the structural foundation facilitates brain dynamics in the infant's brain: (1) whether the infant's brain is controllable and how its capability in controlling state transitions develops during the early stages of infancy; (2) how much control energy is necessary for transitions to different brain functional network activation states and how preterm birth affects the controllability and control energy cost; (3) how NCT explains the distinct cognitive performance at 18 months old. To answer these questions, we investigated the controllability of structural connectomes during infancy using a large sample of 521 infants, including longitudinal data from 73 preterm infants scanned twice across the perinatal period. We characterized the spatial distribution of regional controllability and how controllability develops from 28 weeks gestation through the first month of postnatal life. We simulated the activation of different functional brain networks based on the individual structural connectome and calculated the control energy theoretically required in each

situation. In addition, we investigated the effect of preterm birth on controllability and control energy. Finally, we explored how controllability relates to individual differences in neurodevelopment at 18 months.

## Results

We examined the average and modal controllability of the structural connectomes in 448 term (209 female, 239 male) and 73 preterm (32 female, 41 male) infants from the second data release of the developing Human Connectome Project (dHCP). Term infants were scanned once, at around 2 weeks after birth (mean PMA = 41 weeks). Preterm infants were scanned twice, first at about 2 weeks after birth (mean PMA = 33 weeks) and second at the term-equivalent age (TEA; mean PMA = 41 weeks; Table 1). We used DSI-studio (http://dsi-studio.labsolver.org/) to reconstruct the diffusion data using generalized q-sampling imaging and create structural connectomes with mean quantitative anisotropy value for an infant-specific atlas consisting of 90 nodes (Fig. 1a)[20]. Average and modal controllability were calculated from the structural connectome for each infant (Fig. 1b). As average and modal controllability are not independent in brain networks[13], we focus the results on average controllability. Additional details and results on modal controllability can be found in the supplementary material (Fig. S1).

### Controllability of the infant structural connectome

First, we investigated whole-brain (defined as the mean across all brain regions in a single infant) average and modal controllability in the infant brain. For all three groups, there was a negative correlation between whole-brain average controllability and modal controllability (term: $r = -0.37$, $p = 6.6e-16$; preterm at birth: $r = -0.84$, $p = 8.5e-21$; preterm at TEA: $r = -0.34$, $p = 0.0031$), suggesting that infant brain may prioritize the maturing support for one control strategy over the other through its development (Fig. 2a). These results contrast positive associations between whole-brain average and modal controllability observed in adolescents and adults[13,18]. This may represent a shift in structural brain controllability across development aligned with the developmental trajectory of the brain's morphometric phenotypes[21]. There was no significant difference in these correlations between the term and preterm at TEA groups. However, the correlation between whole-brain average and modal controllability was significantly less negative at TEA than at birth for preterm infants ($z = 5.13$, $p = 3.0e-7$), further suggesting a developmental shift in average and modal controllability in early life. Whole-brain average and modal controllability may start negatively correlated in the third trimester, become less negatively correlated over infancy and toddlerhood, and eventually transition to positively correlated in childhood.

## Table 1 | Demographic characteristics (mean (std))

| Group | | Preterm | | Term |
|---|---|---|---|---|
| Scan | | At birth | At TEA | |
| Sex, num of females (%) | | | 32/73 (43.84%) | 209/448 (46.65%) |
| Gestational Age (GA) at birth (weeks) | | | 30.72 (3.66) | 39.93 (1.26) |
| Post Menstrual Age (PMA) at scan (weeks) | | 33.73 (2.32) | 41.38 (1.58) | 41.19 (1.72) |
| Motion parameters | Average intra-volume translation (mm) | | 0.12 (0.051) | 0.13 (0.052) |
| | Average intra-volume rotation (a.u.) | | 0.18 (0.10) | 0.19 (0.092) |
| | Outlier ratio (%) | | 0.22 (0.088) | 0.25 (0.083) |
| BSID-III | Num of subjects (%) | | 56 (77%) | 356 (79%) |
| | Cognition | | 9.31 (2.95) | 10.10 (2.12) |
| | Language | | 17.91 (6.22) | 19.28 (5.09) |
| | Motor | | 19.26 (4.03) | 20.62 (3.15) |

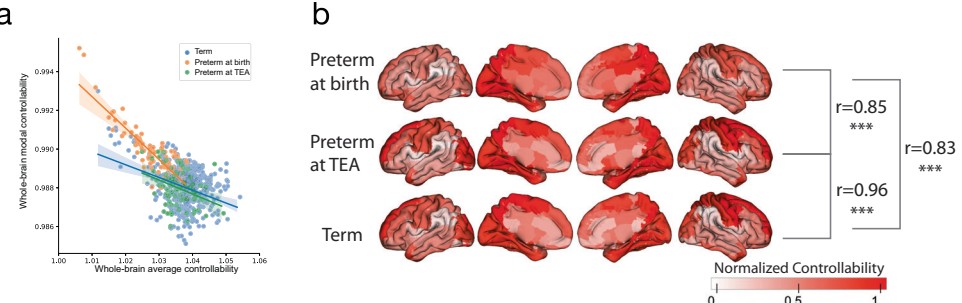

**Fig. 2 | Controllability distribution of the infant's brain. a** Negative associations between whole-brain (or, the mean controllability across every brain region for a single infant) average controllability and whole-brain modal controllability among three subgroups (Pearson's correlation: term $r = -0.37$, $p = 6.6e{-}16$; preterm at birth $r = -0.84$, $p = 8.5e{-}21$; preterm at TEA $r = -0.34$, $p = 0.0031$; two-sided). Each dot represents the whole-brain average and modal controllability for a single infant. The shaded envelope denotes the 95% confidence interval. **b** Normalized regional average controllability was spatially similar across the preterm at birth, preterm at TEA infants, and term groups (Pearson's correlation: preterm at birth vs preterm at TEA: $r = 0.85$, $p = 5.5e{-}26$; preterm at TEA vs term: $r = 0.96$, $p = 1.1e{-}49$; preterm at birth vs term: $r = 0.83$, $p = 4.0e{-}24$; two-sided). (*, **,*** indicates results are significant at $p < 0.05$, $p < 0.01$, and $p < 0.001$ for Pearson's correlation.) Source data are provided as a Source Data file.

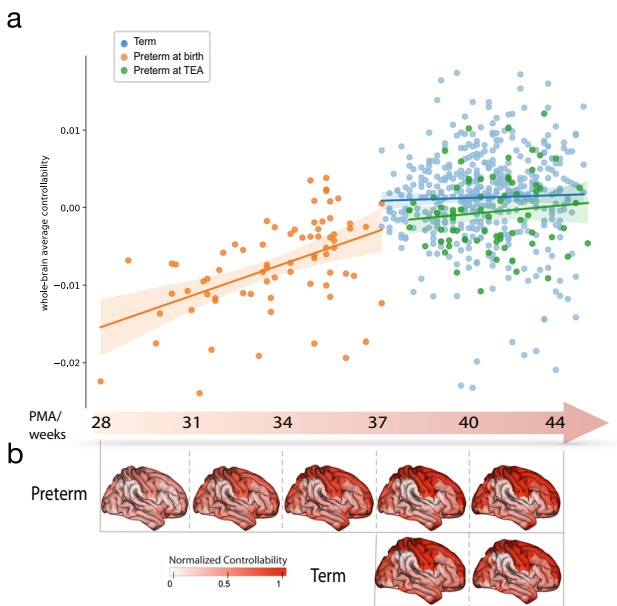

**Fig. 3 | Controllability development during the perinatal period. a** Average controllability changes more rapidly between 28 and 36 weeks and begins to level out around birth (Pearson's correlation: preterm at birth: $r = 0.54$, $p = 9.5e{-}7$; preterm at TEA: $r = 0.11$, $p = 0.36$; term: $r = 0.029$, $p = 0.54$; two-sided). The shaded envelope denotes the 95% confidence interval. **b** The gradual change in normalized average controllability is shown underneath the timeline on the brain maps. Source data are provided as a Source Data file.

The spatial distribution of the brain regions with high average controllability was similar, but not the same, across each group (Fig. 2b). On the network level, regions showing the highest average controllability across groups were primarily located in the visual, dorsal attention, and somatomotor networks. More detailed region information is listed in supplementary materials (Table S1, Fig. S2). Compared to the network roles of the controllability in adult's brain structural networks[13], visual and somatomotor networks play a similar role in average controllability, while the controllability of default mode network appears to be different in the infant brain.

### Controllability development over time
Results from the longitudinal sample of preterm infants showed that the whole-brain average controllability significantly increased over the perinatal period (Fig. 3a). Whole-brain average controllability

developed more rapidly ($z = 2.92$, $p = 0.0017$) between 28 and 36 weeks PMA ($r = 0.54$, $p = 9.5e{-}7$) compared to 38–44 weeks PMA for preterm infants ($r = 0.11$, $p = 0.36$), indicating that the rapid development of average controllability during the third trimester slowed down around birth (Fig. 3a).

Node-wise comparison in the development rate showed that the nodes with significant differences in their changes between the two scans were mainly concentrated in the frontal and occipital lobes (Fig. S3). The rate of controllability development of the frontal lobe at the first scan during 28–36 weeks (mean node-wise $r = 0.44$) was faster than that of the second scan at TEA (mean node-wise $r = 0.23$). A similar situation was observed in the occipital lobe, where the first scan witnessed a relatively rapid increase during 28–36 weeks (mean node-wise $r = 0.49$), while the increment during the second scan at TEA slows down (mean node-wise $r = 0.29$).

For the term group, postmenstrual age at scan negatively correlated with whole-brain modal controllability ($r = -0.50$, $p = 2.7e{-}29$; Fig. S1c), but not whole-brain average controllability ($r = 0.029$, $p = 0.54$; Fig. 3a). Together, these results suggest that the brain may asynchronously prioritize the average controllability over the modal controllability to support the establishment of a fundamentally efficient brain network that is advantageous in driving the brain into many easy-to-reach states in its early developmental stage.

### Preterm birth alters controllability of the infant brain
Term infants exhibited significantly higher whole-brain average controllability compared to preterm infants at TEA ($t = 3.28$, $p = 0.006$, df = 519). In addition, we examined regional differences in average and modal controllability between the term infants and preterm infants at TEA. Compared to term infants, preterm infants at TEA exhibited widespread, weaker average controllability ($p < 0.05$, FDR corrected) in the bilateral temporal lobes, motor cortex, inferior frontal gyrus, precuneus, and cuneus and sparser, greater average controllability in bilateral angular gyrus and orbital frontal cortex (Fig. 4a).

On the whole-brain level, there were no group differences between term and preterm infants at TEA for the average or modal controllability development rate. On the regional level, most preterm-birth-affected brain nodes are concentrated in the frontal lobes, where preterm infants (mean node-wise $r = 0.23$) establish a higher development rate compared to their term fellows (mean node-wise $r = -0.054$) (Fig. 4b). Notably, these affected regions are also included in a larger subset of occipital regions that significantly drive the longitudinal development of infants' brains. Regional controllability development rate distribution for preterm and term infants can be found in Fig. S4.

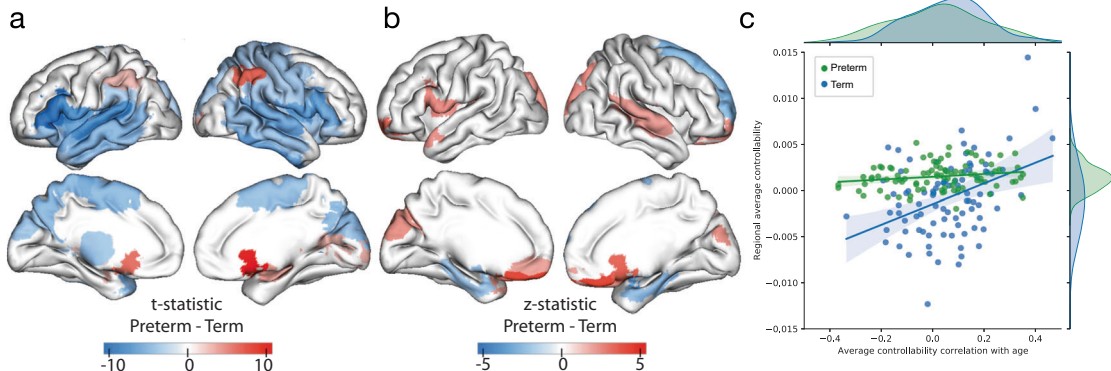

**Fig. 4 | Group differences in controllability between preterm and term infants.** **a** Average controllability distribution differences between preterm and term infants (t stats value from the two-sampled two-sided *t*-test). **b** Average controllability development rate differences between preterm and term infants (z stats value from correlation comparison). **c** Positive correlations between regional average controllability and their rate of development (correlation with age) for both preterm (Pearson's $r = 0.40$, $p = 1.1e-4$; two-sided) and term infants (Pearson's $r = 0.18$, $p = 0.083$; two-sided). The shaded envelope denotes the 95% confidence interval. Source data are provided as a Source Data file.

**Fig. 5 | Control energy cost to activate brain networks.** Whole-brain control energy cost to activate each brain functional network were compared between term (dark box on the left) and preterm infant (light box on the right) groups with an unpaired two-sided *t*-test: Visual: $t = 1.96$, $p = 0.050$; Somatomotor: $t = -4.74$, $p = 2.7e-6$; Dorsal attention: $t = 5.42$, $p = 9.2e-8$; Ventral attention: $t = -3.23$, $p = 0.0010$; Limbic: $t = 2.84$, $p = 0.005$; Frontoparietal: $t = -5.35$, $p = 1.3e-7$; Default mode: $t = 0.41$, $p = 0.68$; Subcortical: $t = -1.83$, p-0.067. Boxes denote the 25th to 75th percentile and the median line. (*, **,*** indicates results are significant at $p < 0.05$, $p < 0.01$, and $p < 0.001$ for the two-sample two-sided *t*-test.) Regional control energy required for term (on the left) and preterm infants (on the right) to reach each network activation target were shown on the corresponding brain maps. Source data are provided as a Source Data file.

Given the whole-brain trend of increasing average controllability and decreasing modal controllability with increasing age, we examined how regional controllability affects its development rate. We found significant correlations between a region's controllability and its development rate for average and modal controllability. In other words, regions with higher values of average controllability exhibited a greater increase in average controllability with age for both term ($r = 0.18$, $p = 0.083$) and preterm at TEA ($r = 0.40$, $p = 1.1e{-}4$) groups. This association was significantly stronger from the preterm infants at TEA than the term infants ($z = 1.88$, $p = 0.030$) (Fig. 4c).

### Sensitivity analyses

To ensure robust results, we conducted several sensitivity analyses. Previous work has noted widespread changes in the structural connectome during the perinatal period[3–8]. We investigated the sensitivity of our results to other basic network features (i.e., strength and network density; Table S2). When controlling for strength and network density, results are similar with interpretations remaining unchanged (see Tables S3, 4 and Figs. S5, 6). In addition, sex differences are known in development and preterm birth. Thus, we repeated analyses for female and male infants separately (Tables S5, 6) to investigate potential sex effects. No significant change in the main results was found due to sex differences.

### Control energy cost to activate canonical functional networks

We further explored how the structural connectome may facilitate brain dynamics in the infant brain. Using a validated simulation approach[15], we modeled state transition from the initial baseline state to target "activated" states for eight canonical resting-state networks, constrained by the structural connectome. At the initial state, the magnitude of all regions' activity was defined as 0 to act as the baseline; for each target state, the magnitude of regions in each functional network (i.e., visual, somatomotor, dorsal attention, ventral attention, limbic, frontoparietal, default mode network; see Table S8) was assigned 1, respectively, to represent different activity patterns. For each simulated state transition, we calculated the control energy cost for each node to drive the desired system state changes for an individual infant based on their structural connectome.

We compared the control energy required by preterm and term infants to activate each functional brain network. At term-equivalent age, term infants required lower energy to activate the frontoparietal ($t = -5.35$, $p = 1.3e{-}7$, df = 518), somatomotor ($t = -4.74$, $p = 2.7e{-}6$, df = 518), and ventral attention networks ($t = -3.23$, $p = 0.0010$, df = 518) than preterm infants. However, preterm infants needed less energy to activate the dorsal attention ($t = 5.42$, $p = 9.2e{-}8$, df = 518) and limbic ($t = 2.84$, $p = 0.005$, df = 518) networks. Interestingly, there were no differences between the two groups in energy expenditure required to activate visual, subcortical, and default mode networks, indicating that preterm birth may not affect basic brain functions related to visual tasks or resting-state.

In general, for each network, the nodes belonging to that network required the largest amount of energy to activate the target network (Fig. 5). Though, for most networks, control energy in other networks was required to activate a network. For example, nodes in the visual network required the largest energy to activate the visual network, but nodes in the dorsal attention network and frontoparietal also exhibited positive control energy. The ventral attention and frontoparietal networks were exceptions in that positive control energy was restricted to nodes in the ventral attention and frontoparietal networks, respectively. Finally, the control energy expenditure of the transition from the baseline state to the targeted state was significantly lower in real brain networks than in null model networks that preserved both the strength and degree distribution (Table S7).

### Controllability at birth is associated with cognitive assessments at later ages

Finally, we investigated if individual differences in controllability were associated with cognitive assessments at 18 months, measured by the Bayley Scales of Infant and Toddler Development, 3rd Edition (BSID-III). This analysis focused on a subset of 412 infants (356 term and 56 preterm infants) that had completed the assessment (mean age = 18.92 months, s.d. = 1.78 months). Significant correlations were found between BSID-III cognitive scores and controllability measurements on the whole-brain level (average controllability: $r = 0.12$, $p = 0.014$; modal controllability: $r = -0.24$, $p = 1.3e{-}6$). Furthermore, from a longitudinal perspective, a positive correlation was observed between the cognitive score and the increasing modal controllability for preterm infants from birth to TEA ($r = 0.31$, $p = 0.023$). Figure S7 shows scatter plots for each of these associations, along with the correlations between controllability and other assessments (i.e., language and motor) from BSID-III.

## Discussion

Leveraging network control theory, we investigated the controllability of structural connectomes in the perinatal period. Elements of controllability were present at the beginning of the third trimester. Average controllability developed earlier compared to modal controllability during the third trimester. At birth, regions with the highest average controllability were in the frontal and occipital lobes. Areas with the highest modal controllability were in the temporal and parietal lobes. These broad patterns of controllability were observable in preterm infants. However, regional controllability was ultimately altered in several cortical regions. In addition, controllability at birth was associated with cognitive ability at 18 months. Together, our results exhibited the development of the controllability of structural connectomes in the perinatal period.

During the perinatal period, the structure of the infant's brain changes rapidly[22–24]. Previous studies on infants have demonstrated that fundamental network topologies of the white matter are present but continue to mature over this period[6,8,25,26]. Further, the structural connectome contains meaningful individual differences that can be used as a 'fingerprint' to identify an individual[9,27] and predict later behaviors[11,28,29]. Consistent with these previous studies, controllability rapidly changes during the newborn period. In addition, we found that brain development is faster in those more controllable regions. This result is consistent with a prior study on controllability development during adolescence[18], showing that stronger average 'controllers' experience a more significant increase from 8 to 20 years old.

The contrasting spatial and developmental patterns of average and modal controllability are consistent across our analyses. These patterns likely reflect the distinct but complementary nature of these measures: brain regions with strong average controllability are mainly located in the default mode system and more active during resting state, while strong regions in modal controllability appear more in the cognitive control system including frontoparietal and cingulo-opercular systems and play important roles in tasks requiring high-level cognitive control or task-switching according to previous observations in the young-adult group[17]. Early in the perinatal period, average controllability develops before modal controllability with rapid changes. Whole-brain average and modal controllability are also inversely correlated, suggesting the infant connectome can only support either nearby or far away state transitions, but not both. These findings contrast with those from older individuals, where average and modal controllability increase simultaneously over childhood and adolescence[18]. Nevertheless, these results are consistent with infants' developmental trajectories of behavior. Newborns first learn sensory and motor behaviors (e.g., crying, grasping, listening)[30–32]. These behaviors rely on similar neural circuits and presumably similar brain states, thus, requiring the brain to support the transition between

nearby states (i.e., higher average controllability). Further, regions of the highest average controllability are in motor and sensory cortices. In contrast, higher modal controllability facilitates an easier transition between "far way" states and supports cognition and executive functions. Corresponding functional networks are not developed in the perinatal period[33,34]. These early foundations of modal controllability appear essential for cognitive development at 18 months. Overall, the contrasting patterns in average and modal controllability likely reflect the comparative needs of the infant's brain to support developmentally appropriate behaviors.

As more complex behaviors develop, one would expect an increase in modal controllability during later infancy and toddlerhood. Nevertheless, further cross-sectional and longitudinal studies in later infancy and toddlerhood are needed to link controllability findings in early infancy and older individuals and to associate these changes with emerging behavior. Studies in this period will allow for investigating individual differences in controllability and behavior, as quantifying behavior in newborns is difficult.

Our analysis suggests that preterm birth subtly alters the controllability of structural connections. Consistent with these observations, altered structural connectivity is widely reported in preterm birth[8,35-38]. While similar patterns of controllability remained intact, we show evidence of alteration in both average and model controllability for preterm infants. Intriguingly, many regions exhibiting these alterations are in regions underlying well-known behavioral deficits in individuals born preterm. For example, group differences in controllability are observed in Broca's and Wernicke's areas. Language deficits are common in preterms, with functional differences observable from the perinatal period through adulthood[39-41]. Similarly, altered controllability in the motor cortex and social processing regions are consistent with differences observed in preterm infants, children, and adolescents[42,43]. Altered controllability in the perinatal period could provide a structural explanation for these later functional differences.

While preterm infants show differences in controllability at TEA, they might be able to "catch-up" to their term-born peers. The slopes of the regional developmental for preterm infants are steeper than those of term infants, suggesting that the regional controllability of preterm infants at TEA matures faster than that of term infants. In general, catch-up growth is typical in preterm infants and children. For example, preterm infants weigh significantly less than their term-born peers, but these differences are smaller or disappear at older ages[44-46]. Additional studies in school-age children, adolescents, and adults born preterm will be needed to map out these developmental trajectories and assess if changes in controllability persist later in life.

Based on the control energy results, the white matter architecture that supports future functional networks appears to be in place before more complex behaviors emerge. For all networks, the control energy of brain state transition was lower than in null models. This result aligns with the prior study of control energy in the adolescent group[15]. It indicates that brain topology is constructed to support efficient brain dynamics even in early infancy. Finally, a mix of increased and decreased control energy was observed in preterms compared to term infants. Term infants were more efficient in facilitating the somatomotor and frontoparietal networks, which may be attributed to the rapid development of interhemispheric connectivity of these networks during the perinatal period[47]. Further follow-up data is needed to understand how these differences potentially mediate development outcomes in preterms.

Several limitations should be noted. First, the term data is cross-sectional from a relatively narrow age range. The longitudinal data across the perinatal period is only for preterm infants. The lack of longitudinal data for term infants and a broader developmental period prevents inferences about the within-subject developmental trajectories beyond the newborn period. As such, how controllability develops through later infancy and toddlerhood remains unknown, as

how these changes support cognitive development. Follow-up longitudinal studies will be needed to clarify how and when controllability reaches similar patterns to those observed in school-age children and how these relate to cognitive outcomes. Once these trajectories are known, future work could investigate how modified environment factors can lead to changes in these trajectories. Second, though the sample size is large for infant neuroimaging studies, preterm birth is highly heterogeneous in its causes and outcomes[48]. The size and variability of the preterm birth effects may not reflect a broader population of preterm infants. Third, the linear model used in the network control theory remains limited in fully unraveling brain dynamics. However, both linear and nonlinear models can be used to successfully predict brain dynamics from brain structure[49]. Also, the current method is state-of-art in resolving the complex structural features of human brain networks[50]. Similar methods have achieved remarkable results in understanding healthy brain functions and targeting therapeutic interventions for patients[19,51,52]. Fourth, consistent with our controllability studies, we used simulations on a simplified model to associate controllability with changes in brain dynamics, instead of real fMRI data. Establishing associations with real fMRI data remains a challenging but needed next step. Finally, several demographic and scanning-related factors, including sex, brain volumes, motion artifact, and other signal noise during scanning, can bias the estimation of developmental trajectories. While we conduct the age-related analysis with the above factors as covariances to limit the impact of these confounding variables, they may still influence the results and would benefit from further study.

In conclusion, we investigated the developmental trajectories of the controllability of the infant structural connectome. Our findings show that the infant brain's controllability changes rapidly during the prenatal period and is altered in preterm birth. This framework can be used as a baseline for the controllability distribution of the infant brain and to further understand the developmental substrates of neurodevelopmental disorders. Insights into the normal network topology and the alterations associated with neurodevelopmental disorders may lay the foundation for individualized treatment to correct developmental trajectories and improve outcomes.

## Methods
### Participants
All data were acquired from the Developing Human Connectome Project[53] (dHCP, http://www.developingconnectome.org/), a large, cross-sectional open science study of infant brain development. The study was approved by the National Research Ethics Service West London committee, and written consent was obtained from participating families before imaging. We included 448 term infants (209 female, 239 male) and 73 preterm infants (32 female, 41 male) with longitudinal scans (at birth and TEA) from the second data release of dHCP. Preterm infants were born between 23.57 weeks and 37.00 weeks of gestation and scanned twice. The first was around 3 weeks after birth, at a mean of 33.73 weeks. The second was at term-equivalent age with a mean of 41.38 weeks. Term infants were born between 37.14 weeks and 42.29 weeks of gestation and scanned between 37.43 weeks and 44.71 weeks. Demographic information is summarized in Table 1.

### MRI acquisition
All scans were collected in the Evelina Newborn Imaging Centre, St Thomas' Hospital, London, UK. Diffusion magnetic resonance imaging (MRI) data and all other MRI data were acquired with a Philips Achieva 3 T scanner (Philips Medical Systems, Best, The Netherlands) with a dHCP-customized neonatal imaging system including a 32-channel receive neonatal heal coil (Rapid Biomedical GmbH, Rimpar, DE)[53]. Infants were scanned during unsedated sleep after feeding and immobilization in a vacuum-evacuated bag, with hearing protection

 

and physiological monitoring (including pulse oximetry, body temperature, and electrocardiography data) applied during scanning.

T2-weighted images were obtained using a Turbo spin echo sequence (TR = 12 s, TE = 156 ms, SENSE factor 2.11 (axial) and 2.54 (sagittal) with overlapping slices (resolution = $0.8 \times 0.8 \times 1.6$ mm$^3$). T2w images were motion corrected and super-resolved to a resolution of $0.8 \times 0.8 \times 0.8$ mm$^3$ [54]. Diffusion-weighted imaging (DWI) was obtained in 300 directions (TR = 3.8 s, TE = 90 ms, SENSE factor 1.2, multiband factor 4, and resolution $1.5 \times 1.5 \times 3$ mm$^3$ with 1.5 mm slice overlap) with b-values of 400 s/mm2, 1000 s/mm$^2$ and 2600 s/mm$^2$ spherically distributed in 64, 88 and 128 directions respectively using interleaved phase encoding.

Diffusion MRI was reconstructed at an in-plane resolution of 1.5 mm and slice thickness of 1.5 mm. Images were denoised and corrected for motion, eddy current, Gibbs ringing, and susceptibility artifact with the diffusion SHARD pipeline. In-scanner head motion was estimated by the SHARD outlier ratio, which is the mean outlier weight of all slices detected in slice-to-volume reconstruction. A quality check was conducted by neighboring DWI correction (NDC)[55], leading to 34 scans being excluded due to their low NDC values calculated by a median value-based outlier detector. The accuracy of b-table orientation was examined by comparing fiber orientations with those of a population-averaged template[56]. Then, the reconstruction of the diffusion data was performed in native space with generalized q-sampling imaging (GQI) (Yeh, Wedeen, and Tseng 2010) with a diffusion sampling length ratio of 1.25. The tensor metrics were calculated and analyzed using the resource allocation (TG-CIS200026) at Extreme Science and Engineering Discovery Environment (XSEDE) resources[57].

After reconstructing images with GQI, the whole-brain fiber tracking was conducted with DSI-studio (http://dsi-studio.labsolver.org/) with quantitative anisotropy (QA) as the termination threshold. QA values were computed in each voxel in their native space for every subject. The tracking parameters were set as the angular cutoff of 60 degrees, step size of 1.0 mm, minimum length of 30 mm, and maximum length of 300 mm. The whole-brain fiber tracking process was performed with the FACT algorithm until 1,000,000 streamlines were reconstructed for each individual. Here, we used a neonatal AAL-aligned brain parcellation with 90 nodes[20] to construct the structural connectome for each infant. T2-weighted images in native DWI space were used to provide information on region segmentation during the construction of connectomes. The structural connectome for each individual was then constructed with a connectivity threshold of 0.001 and the pairwise connectivity strength was calculated as the average QA value of each fiber connecting the two end regions, which results in a $90 \times 90$ adjacent matrix for each participant as the brain structural connectome matrix.

## Network control theory

The human brain can be considered as a natural complex system. One important way to understand the behavior of a complex system is from the mechanism to control it, which means driving the system to the desired state. Network control theory, aimed to address the problem of how to control a system consisting of nodes (brain regions) and edges (white matter tracts between brain regions), can illustrate the dynamic changes in the short term or the developmental trajectory in the long term that human brain goes through and distinguish the cognitive dysfunction in disorder groups.

A network system can be represented as the graph $\mathbf{G} = (\mathbf{V}, \mathbf{E})$, where $\mathbf{V}$ and $\mathbf{E}$ are the vertex and edge sets respectively. Let $a_{ij}$ be the weight associated with the edge $(i,j)$ in $\mathbf{E}$ and define the weighted adjacency matrix of the graph $\mathbf{G}$ as $\mathbf{A} = [a_{ij}]$, where $a_{ij} = 0$ when $a_{ij} \notin \mathbf{E}$. Here, the individual structural connectome $\mathbf{A}$ in $\mathbb{R}^{n \times n}$ is a symmetric and weighted adjacency matrix whose elements $[a_{ij}]$ evaluate the strength of the white matter fiber connecting between region $i$ and region $j$ in the brain.

## Dynamic processes and controllability metrics

For the definition of the neural dynamic processes, we adopt the prior models that link brain structural networks to simplified brain dynamics. Though the evolution of brain activity occurs in a nonlinear manner, previous studies have demonstrated that the simplified linear models can predict a significant portion of the variance in the neural dynamic recorded by fMRI. Therefore, we employ a simplified noise-free linear discrete-time and time-invariant network model: $\mathbf{x}(t+1) = \mathbf{A}\mathbf{x}(t) + \mathbf{B}_K(t)\mathbf{u}_K(t)$, where $x$ denotes the brain state at a given time, and $\mathbf{A}$ is the symmetric, undirected and weighted adjacency matrix for the network. In our case, $\mathbf{A}$ represents the structural connectome for each individual, whose element indicates the pairwise strength of brain structural connection and diagonal elements equal to zero. The input matrix $\mathbf{B}_K$ identifies the control points $K$ in the brain, where $K = \{k_1, \cdots, k_m\}$ and $\mathbf{B}_K = [e_{k1} \cdots e_{km}]$. $e_i$ denotes the $i$th canonical vector of dimension $N$ and input $u_K$ denotes the input control strategy over time.

We further study the ability of a certain brain region to drive the state of a dynamic system to a desired state, which is defined as controllability. Classic control theory provides that the controllability of the network from the set of network nodes $K$ is equivalent to the controllability Gramian $\mathbf{W}_K$ being invertible, where $\mathbf{W}_K = \sum_{t=0}^{\infty} \mathbf{A}^\tau \mathbf{B}_K \mathbf{B}_K^T \mathbf{A}^\tau$. The input matrix $\mathbf{B}$ reduces to a one-dimensional vector as we choose control nodes one at a time. Based on this network control theory framework, we examine two diagnostics of controllability that describe the ability to drive the network with different types of transitions as patterns of regional activity: average controllability to measure the ability to drive nearby brain state transition and modal controllability to estimate that of distant brain state transition on the brain energy landscape. Average controllability is defined by the average input energy from a group of control nodes and overall potential target states. The average input energy is proportional to Trace($\mathbf{W}_K^{-1}$), the trace of the inverse of the controllability Gramian. Here, to keep the consistency with previous studies[13,18], we use Trace($\mathbf{W}_K$) as the measurement of *average controllability* to increase the accuracy of computation on small brain networks and maintain the information obtained by this measurement. As noted in the prior study, Trace($\mathbf{W}_K$) encodes a well-defined metric for controllability, equivalent to the network $H_2$ norm or the energy of the network impulse response. *Modal controllability* is defined as the ability of a node to control difficult-to-reach mode of the dynamic network system and is computed from the eigenvectors $[v_{ij}]$ of the adjacency matrix $\mathbf{A}$. Here, we define the measurement of modal controllability as $\phi_i = \sum_{j=1}^{N}(1 - \lambda_j^2(\mathbf{A}))v_{ij}^2$ of all N modes $\lambda_0(\mathbf{A}), \ldots, \lambda_{N-1}(\mathbf{A})$ from brain region $i$, following the definition of previous studies.

## Control energy cost during state transitions

To explore how the brain's dynamic processes are constrained by the structural connectome, we utilized network control theory to model the energy required to activate specific brain networks over the existing structural network topology of the infant brain. The baseline state $\mathbf{x}(0)$ was set to zero to simulate the resting state of the brain, while the target brain state $\mathbf{x}(T)$ was defined such that all regions in the desired brain network had a magnitude of one, while all other regions had a magnitude of zero, representing activation of the desired regions.

Here, we followed the definition of the control task in the previous studies[15,58], where the system transitions from initial state to target state with minimum-energy input as an optimal control problem, with the cost function defined as a Hamiltonian, $H(\mathbf{p}, \mathbf{x}, \mathbf{u}, \mathbf{t}) = \mathbf{x}^T\mathbf{x} + \mathbf{u}^T\mathbf{u} + \mathbf{p}^T(A\mathbf{x} + \mathbf{B}\mathbf{u})$. By solving the optimization problem $\mathbf{u}^* = arg\ min(H)$, control energy for each node $k_i$ was defined as $E_{k_i} = \int_{t=0}^{T}||\mathbf{u}_{k_i}^*(t)||^2 dt$, indicating the overall energy input required by the node to facilitate the desired state transition.

To validate whether the structure of infant brain networks facilitated transitions between the baseline state and targeted brain states, we constructed a null model network with the preserved degree and strength distribution of the original network for each individual[59] and compared the energy consumption differences between the null model network and the original network with a paired two-sided t-test.

## Statistical analysis

Whole-brain controllability was calculated as the mean controllability across all brain regions, and regional controllability was calculated as the mean controllability across all subjects. Confounds including sex, brain volume, head motion, network strength and network density were included in statistical analyses as additional regressors. To examine whether significant differences exist in the regional controllability between preterm infants at TEA and term infants, a two-sample two-sided t-test was performed. The Benjamini-Hochberg false discovery rate (FDR) method was used to correct for multiple comparisons. Pearson correlations were used to associate average and modal controllability on the whole-brain and regional level and postmenstrual age and controllability. To compare whether the association between controllability and age differs across different groups, a comprehensive statistical comparison of correlations with cocor() function in cocor package in R version 4.1.1[60] was used for the comparison of whole-brain controllability development rate between preterm at birth and preterm at TEA, and for the comparison of regional controllability development rate between preterm at TEA and term. The correlation comparison test with cocor package generated a z-statistic to indicate the direction and amplitude of the difference between two correlations.

## Reporting summary

Further information on research design is available in the Nature Portfolio Reporting Summary linked to this article.

## Data availability

Raw Data from the Developing Human Connectome Project is publicly available at http://www.developingconnectome.org/data-release/third-data-release and can be downloaded upon request from https://biomedia.github.io/dHCP-release-notes/download.html. The relevant data to generate the figures are provided as Source Data files. Source data are provided with this paper.

## Code availability

Preprocessing code can be found at https://brain.labsolver.org/hcp_d2.html. Network control theory code can be found at https://complexsystemsupenn.com/codedata: (controllability https://complexsystemsupenn.com/s/controllability_code-smb8.zip; control energy https://github.com/ursbraun/network_control_and_dopamine). Custom analysis code is available at https://github.com/huiliii/infant_control.

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

## Acknowledgements

This work is supported by NIH(R01MH126133). Data were provided by the developing Human Connectome Project, KCL-Imperial-Oxford Consortium funded by the European Research Council under the European Union Seventh Framework Programme (FP/2007-2013)/ERC Grant Agreement no. [319456]. We are grateful to the families who generously supported this trial.

## Author contributions

H.S.: conceptualization, methodology, investigation, result visualization, and writing-original draft. R.J. and W.D.: investigation and result visualization. A.D. and S.G.: conceptualization, methodology and writing-review and editing; S.N. and M.S.: writing-review and editing; D.S.: conceptualization, supervision and writing-review and editing.

## Competing interests

The authors declare no competing interests.
