## [Peer Review File · Nature Communications]

Network controllability of structural connectomes in the neonatal brainReviewer #1 (Remarks to the Author):

In this manuscript Sun and colleagues study the developing structural connectome using network controllability theory (NCT). This is an interesting and valuable study making good use of a large sample of neonatal MRI publicly available as part of the Developing Human Connectome Project (dHCP). The results presented are relevant, characterising the maturation of controllability features during early development, how preterm birth impacts network controllability, and the association of controllability at birth with cognitive ability at 18 months. I find this manuscript is written well and it is easy to follow, however I think there are a few methodological issues that should be clarified, and I have a few suggestions for the authors, which I think would improve their study:

-The authors study structural connectome controllability, and argue the value of NCT to answer the question: “how does an infant’s structural connectome develop to constrain later brain dynamics?”. However, they do not use resting-state data (available as part of the dHCP for the same subjects) to assess the association of NCT features with functional connectivity in neonates. I am not sure how the authors justify adopting “prior models that link brain structural networks to resting-state functional dynamics” without assessing how well these (linear) models actually fit resting-state functional dynamics in neonates. It would be of course a nice addition to the manuscript if the authors were to test the capacity of NCT on structural connectivity to predict brain dynamics in neonates, perhaps in a subset of the sample, or using publicly available derivative data from other dHCP studies. If the authors consider this to be out of the scope for their study, I suggest they justify why adapting NCT to neonatal function is not required, or discuss whether this is a limitation of their study.

-I wonder what is the effect of basic network properties on NCT features, i.e., is controllability developing “independently”, or is it influenced by the general development of structural connectivity in neonates (e.g., increased WM volume and FA/QA)? The authors partially address this issue in their supplementary materials (though not in Table S2 as stated in the text, Results P8), where they assess the use of average strength as a covariate for the association of NCT features with PMA (but not for other statistical associations, e.g., term vs preterm). I think it would be a nice addition to also assess the effect of network density (in addition to average strength), and to directly characterise the correlation of basic network features (average strength and network density) with the NCT features reported. Is the strength of this (potential) relationship different between term and preterm? I would also suggest adding descriptive statistics of average strength and density to Table 1.

Other comments

Results:

- P5. Please state the metric used to infer connectivity between ROIs from tractography (e.g., average/median QA).
- Could the authors provide exact p-values when $p < 0.05$ (and $p > 0.001$)?
- Did the authors assess associations with other BSID-III domains (motor and language)? If not significant, please report. Correlations with cognitive scores are relatively weak but significant: for transparency, I would recommend adding scatter plots as supplementary figures.

Methods:

- Please clarify how was QA between ROIs obtained to construct structural connectomes. Was an average/median QA value obtained for each streamline, and then averaged for all streamlines connecting two ROIs? Or average/median QA_0 averaged for all coordinates sampled by the streamlines connecting two ROIs? Please justify the use of mean or median statistics.
- Were subjects with major lesions in WM excluded? Any other exclusion criteria (e.g. motion)?
- How was head motion calculated? Did the authors use SHARD motion estimates? Was there an association between motion estimates and controllability features? Could descriptive statistics of motion be added to Table 1?

Reviewer #2 (Remarks to the Author):

Review of Sun, Jiang et al
Controllability of structural connectomes in the neonatal brain

This manuscript takes an intriguing approach, of using network control theory to investigate the controllability of networks in the infant brain, building on the dHCP open dataset. This seems very interesting and would be a novel and important contribution.

However, at present, I have a serious concern about the validity of the analyses. My concerns were raised by the results shown in Figure 2B. It is vanishingly rare in MRI that any two independent measures correlate with $|r| > 0.99$. This is particularly the case for infant MRI data, which are typically of lower signal-to-noise than adult data. My concern was that the two measures are not independent in some way and so the correlation was being inflated. To investigate this, I used the author's code, which they helpfully provided. To investigate the possibility that the average and modal controllability measures are not independent, I generated a random connectivity matrix (here for a notional 50 nodes and 100 subjects). I then ran the first part of the analysis and graphed the output (code attached in Appendix A below).

Even for these completely random data, there is a correlation across nodes, when averaged by subject ($r = -0.993$, $p < 0.001$, scatterplots attached). This shows that the modal and average controllability are not independent. I could not verify that this would be true for random data with the same distribution as the true connectivity, as this was not provided. However, a similar strong correlation persists if the data are made more sparse (by raising the uniform distribution random values to the power of six) or are changed in range from 0-1 to 0-100.

It would be helpful for the authors to consider this issue, and to check the evaluation code in Appendix A. I have not investigated the validity of the later results, but it would be helpful for the authors to also review these with similar concern in mind.

APPENDIX A

```
addpath(fullfile(pwd,'controllability_code'))
% Make up random data for testing
A_subn=rand(50,50,100);
[node_num,~,sub_num] = size(A_subn);
%% controllability
aver_cons = zeros(node_num,sub_num);
for s = 1:sub_num
aver_cons(:,s) = ave_control(A_subn(:,:,s));
end
modal_cons = zeros(node_num,sub_num);
for s = 1:sub_num
modal_cons(:,s) = modal_control(A_subn(:,:,s));
end
%% aver vs modal
% across subjects
[rs,ps] = corr(mean(aver_cons)',mean(modal_cons'))
% across region
[rr,pr] = corr(mean(aver_cons,2),mean(modal_cons,2))
figure(1);
subplot(211)
scatter(mean(aver_cons)', mean(modal_cons'))
xlabel('Average control')
ylabel('Modal control')
title(sprintf('By subject averaged across nodes r=%.3f, p=%.3f',rs,ps))
subplot(212)
scatter(mean(aver_cons,2), mean(modal_cons,2))
xlabel('Average control')
```

```
ylabel('Modal control')  
title(sprintf('By node averaged across subjects r=%.3f, p=%.3f',rr,pr))
```

Reviewer #2 Attachment on the following page

By subject averaged across nodes $r=-0.320$, $p=0.001$

By node averaged across subjects $r=-0.993$, $p=0.000$

Reviewer #3 (Remarks to the Author):

Here the authors present an analysis of publicly available connectome data from infants at various gestational ages and analyse them through the now well-characterised controllability framework, focussing on two controllability measures - average and modal controllability.

They then provide descriptive statistics of group differences and longitudinal changes, which are interesting. From the current version, however, it is not clear whether these facilitate any novel insights, since no specific hypotheses are given that were a priori setting out to test specific predictions. Furthermore, little is done statistically to account for a whole host of other changes that occur in networks during this developmental time period to identify which (if any) of the described changes are specific to 'controllability' and how much of these are 'just' corollaries to other developmental changes?

Major comments

- The controllability measured discussed in this paper are quantitative network features, similar to others previously discussed in the infant brain. The author's main hypothesis was that 'controllability developed rapidly across the perinatal period and infants born preterm will have altered controllability' - hypothesis is almost certainly going to be true, given what we already know about changes in network topology during development. It is unclear which aspect of the hypothesis testing specifically leveraged controllability vs just applied controllability as an interesting novel metric from a dataset that was on hand.

- Related to this, the paper as currently presented does not fully test how specific any of its findings are to controllability, and how many may be shaped by e.g. changes in weighted degree or other network measures. This is particularly relevant to aspects where additional inference is drawn - e.g. Fig. 5 - the text here implies that the authors wanted to test 'how regional controllability affects its maturation rate'. The statistics presented don't demonstrate this conclusively, as the reader is not given the statistical tools to see whether the observed could be explained away by e.g. weighted degree as network topological feature that might affect both controllability and regional maturation.

Similarly statistical inference across regions should account null models to evaluate which changes are 'specific' to controllability, and which ones are corollaries of wider network changes that would occur in any network with the observed changes in e.g. weighted degree. (e.g. see here: <https://www.ncbi.nlm.nih.gov/pmc/articles/PMC7734595/> for examples of null models that might be considered)

Minor comments

- developmental description - many aspects of brain anatomy change over the first few decades of life; it is therefore not clear what the authors mean by aspects 'maturing by 40 weeks of gestation' (p1)

- wording: control energy is 'biologically evidenced by gray matter integrity and glucose metabolism' - unclear wording

- Figure 2 - not clear to the reader what the difference between A and B is - one is 'whole-brain' (but really just regions averaged by infant); the other one is 'regional' but contains approximately as many dots as plot A? Should there not be more if each region is plotted individually? Are these averaged within region but across infant? (I understand from the methods that this is in fact what has been done, but not clear from the figure alone)

- the inference on infant behaviour is somewhat tricky to relate to the controllability data, and perhaps suggestions about how these could be leveraged into specific testable hypotheses would be helpful

We thank the reviewers and editor for their careful reading of the paper and thoughtful comments. We considered each comment carefully and made extensive revisions, which are believed to significantly improve the clarity and overall strength of the manuscript. For the editor's and reviewers' convenience, we provide a point-by-point response below with the reviewers' comments in black italic and our responses in dark cyan.

REVIEWER #1

In this manuscript Sun and colleagues study the developing structural connectome using network controllability theory (NCT). This is an interesting and valuable study making good use of a large sample of neonatal MRI publicly available as part of the Developing Human Connectome Project (dHCP). The results presented are relevant, characterising the maturation of controllability features during early development, how preterm birth impacts network controllability, and the association of controllability at birth with cognitive ability at 18 months. I find this manuscript is written well and it is easy to follow, however I think there are a few methodological issues that should be clarified, and I have a few suggestions for the authors, which I think would improve their study:

Response:

We are grateful for the reviewer's positive comments and careful reading of the paper.

Major comment #1

-The authors study structural connectome controllability, and argue the value of NCT to answer the question: "how does an infant's structural connectome develop to constrain later brain dynamics?". However, they do not use resting-state data (available as part of the dHCP for the same subjects) to assess the association of NCT features with functional connectivity in neonates. I am not sure how the authors justify adopting "prior models that link brain structural networks to resting-state functional dynamics" without assessing how well these (linear) models actually fit resting-state functional dynamics in neonates. It would be of course a nice addition to the manuscript if the authors were to test the capacity of NCT on structural connectivity to predict brain dynamics in neonates, perhaps in a subset of the sample, or using publicly available derivative data from other dHCP studies. If the authors consider this to be out of the scope for their study, I suggest they justify why adapting NCT to neonatal function is not required, or discuss whether this is a limitation of their study.

Response:

We apologize for the lack of clarity. Previous studies utilizing NCT have only begun to relate NCT to resting-state dynamics. Given the complexity of this task, a majority of the NCT works[1, 2] use a simplified framework to approximate these dynamics by simulating the energy needed to move between a set of representative states. These simulations assume that the brain starts at a neutral state (i.e., no activations in any nodes or canonical brain network) and measures the needed energy to move to a different state, where all nodes in a canonical brain network (e.g., the default mode network) are activated.

In order to further investigate the association between brain activity and structural network architecture during development in infancy, we have added these simulations to our manuscript. We calculated the energy consumption necessary to activate all nodes in 8 different brain networks (Visual, somatomotor, dorsal attention, ventral attention, limbic, frontoparietal, default mode, and subcortical) using NCT[3] and compared between the term and preterm infants.

Our findings indicate that term infants exhibit greater energy efficiency in motor and cognition-related tasks than preterm infants at TEA, as evidenced by their lower energy requirements to activate somatomotor ($t=-4.75$, $p<0.001$, $df=518$, $sd=5.36$), ventral attention ($t=-3.23$, $df=518$, $sd=2.74$, $p=0.001$), and frontoparietal ($t=-5.35$, $p<0.001$, $df=518$, $sd=4.10$) networks. Interestingly, preterm infants at TEA require less energy than term infants to activate the dorsal attention ($t=5.42$, $p<0.001$, $df=518$, $sd=4.79$) and limbic ($t=2.84$, $p=0.005$, $df=518$, $sd=3.35$) networks. Notably, no significant differences were observed between the groups in energy expenditure required to activate visual, subcortical, and default mode network e. These results are now presented in Figure 5 of the main text.

We have removed the phrase “*prior models that link brain structural networks to resting-state functional dynamics*” as it may misrepresent previous work and have added that association with real resting-state data is a needed future direction of research. We have also added this to the limitations section: “Fourth, consistent with our controllability studies, we used simulations on a simplified model to associate controllability with changes in brain dynamics, instead of real fMRI data. Establishing associations with real fMRI data remains a challenging but needed next step.”

Fig. 5. Control energy cost to activate brain networks. Boxplots for control energy averaged over the whole-brain to activate each brain functional network (dark box for term infants, light box for preterm infants). Brain figures for regional control energy required for term (on the left) and preterm infants (on the right) to reach each network activation target.

Major comment #2

-I wonder what is the effect of basic network properties on NCT features, i.e., is controllability developing “independently”, or is it influenced by the general development of structural connectivity in neonates (e.g., increased WM volume and FA/QA)? The authors partially address this issue in their supplementary materials (though not in Table S2 as stated in the text, Results P8), where they assess the use of average strength as a covariate for the association of NCT features with PMA (but not for other statistical associations, e.g., term vs preterm). I think it would be a nice addition to also assess the effect of network density (in addition to average strength), and to directly characterise the correlation of basic network features (average strength and network density) with the NCT features reported. Is the strength of this (potential) relationship different between term and preterm? I would also suggest adding descriptive statistics of average strength and density to Table 1.

Response:

The reviewer brings up a critical point about the specificity of our results to controllability compared to other measures of network topology. We have taken several steps to investigate the specificity of our results. 1) As suggested by the reviewer, we have included network density as a covariate in our analyses. As shown in Table S3 and S4 (see below), controlling for network strength or network density does not meaningfully change our results and conclusions. 2) We have included network density and network strength as covariates in the additional analyses comparing term to preterms (see Fig. S5 and Fig. S6). 3) We have included the descriptive statistics of network strength and network density in Table S2. Analyses detailed here are presented in the supplementary materials, “Controlling for network measurements”).

Table S2: Structural network properties (mean(std))

Group	Preterm		Term
	At birth	At TEA	
Network Strength	0.045(0.015)	0.060(0.0088)	0.065(0.011)
Network Density	0.53(0.15)	0.55(0.070)	0.57(0.069)

Table S3: Controllability maturation during perinatal period controlling for network strength/network density (r-value between controllability and postmenstrual age)

Average Controllability		Modal Controllability	
Controlling for sex, brain volume and head motion (shown in the main text)			
Preterm at birth	0.54***		-0.44***
Preterm at TEA	0.11		-0.35**
Term	0.03		-0.50***
Additionally controlling for network strength			
Preterm at birth	0.59***		-0.14
Preterm at TEA	0.37**		-0.35**
Term	0.54***		-0.53***
Additionally controlling for network density			
Preterm at birth	0.09		-0.23
Preterm at TEA	0.20*		-0.16
Term	0.21***		-0.21***

Table S4: Correlation between regional controllability value and its maturation controlling for network strength/network density (r-value)

Average Controllability		Modal Controllability	
Controlling for sex, brain volume and head motion (shown in the main text)			
Preterm at TEA	0.75***		0.5,***
Term	0.29**		0.3**
z-stats	5.2,***		1.12
Additionally controlling for Network Strength			
Preterm at TEA	0.68***		0.1***
Term	0.2,		0.33**
z-stats	.50***		0.68
Additionally controlling for Network Density			
Preterm at TEA	0.7,***		0.52***
Term	0.25*		0.37***
z-stats	5,***		1.6

A Regional controllability

B Regional controllability maturation rate

Fig. S5: Group differences in controllability between preterm and term infants with network strength as the additional covariance.

A Regional controllability

B Regional controllability maturation rate

Fig. S6: Group differences in controllability between preterm and term infants with network density as the additional covariance.

On the whole brain level, the association between network characteristics such as network strength (weighted degree, or sum of all edge weights) or network density (sum of non-zero edges) and controllability values present similar patterns for term and preterm infants at TEA (Table below). The altered patterns between network characteristics and controllability shown in preterm infants at birth may be caused by the rapid growth of nerve fibers and the unique development pattern of neural myelination at the end of the third trimester.

	Preterm at birth	Preterm at TEA	Term
Corr(Network Strength, AC)	0.94***	0.85***	0.91***
Corr(Network Strength, MC)	-0.73***	-0.03	-0.15**
Corr(Network Density, AC)	0.67***	0.14	0.21***
Corr(Network Density, MC)	-0.43***	0.76***	0.69***

Overall, our results suggest that neither network strength nor density can fully account for the observed developmental trends in controllability in the structural brain networks during early infancy. We have included a new section in the results, labeled “Sensitivity analyses” as well as three new supplemental tables and two new supplemental figures.

Minor comment #1

- P5. Please state the metric used to infer connectivity between ROIs from tractography (e.g., average/median QA).

Response:

Thanks to the reviewer for pointing this out. We used mean QA and have added a clarification in the first paragraph in the Results section:

Page #5: “We used DSI-studio (<http://dsi-studio.labsolver.org/>) to reconstruct the diffusion data using generalized q-sampling imaging and create structural connectomes with mean quantitative anisotropy value for an infant-specific atlas consisting of 90 nodes.”

Minor comment #2

- Could the authors provide exact p-values when $p < 0.05$ (and $p > 0.001$)?

Response:

Sorry for the lack of clarity. In response to the reviewer, all exact p-values have been updated throughout the main text and supplementary materials. The one exception is for the imaging results, where we report as $p < 0.05$, FDR corrected at Page #9, where the field of neuroimaging typically reports threshold maps in this manner (as opposed to listing the exact p-values for a wide range of regions).

Minor comment #3

- Did the authors assess associations with other BSID-III domains (motor and language)? If not significant, please report. Correlations with cognitive scores are relatively weak but significant: for transparency, I would recommend adding scatter plots as supplementary figures.

Response:

We are grateful to the reviewer for bringing this important issue to our attention. We analyzed associations between controllability at birth and BSID-III cognitive, language, and motor scores at 18 months respectively, and updated the results in the main text. For better visualization, we also provided scatter plots in the supplementary material (Fig. S7; also shown below for convenience). Apart from what we have reported in the main text about associations with cognitive performance, individual average controllability was significantly correlated with BSID-III motor scores; individual modal controllability was significantly correlated with both **BSID-III** language and motor scores. However, the correlation between BSID-III scores and increasing modal controllability from the longitudinal perspective did not reach significance in cognitive performance.

Fig. S7: Scatterplots of controllability at birth and behavioral performances at 18 months old.

Significant correlations were observed in BSID-III cognitive scores on the 1st row and individual average controllability ($r=0.12$, $p=0.014$) and modal controllability ($r=-0.24$, $p<0.001$) on the whole-brain level; Increasing modal controllability for preterm infants from birth to TEA was positively correlated with the BSID-III cognitive scores ($r=0.31$, $p=0.023$) (1st row). Individual modal controllability was significantly correlated with BSID-III language on the 2nd row ($r=-0.18$, $p<0.001$) but not average controllability ($r=0.071$, $p=0.15$) and longitudinal changes in modal controllability. BSID-III motor scores on the 3rd row were significantly associated with both individual average ($r=0.14$, $p=0.006$) and modal controllability ($r=-0.18$, $p<0.001$), but not with the longitudinal changes for preterm infants.

Minor comment #4

-Please clarify how was QA between ROIs obtained to construct structural connectomes. Was an average/median QA value obtained for each streamline, and then averaged for all streamlines connecting two ROIs? Or average/median QA_0 averaged for all coordinates sampled by the streamlines connecting two ROIs? Please justify the use of mean or median statistics.

Response:

Sorry for the lack of clarity. To construct structural connectomes using DSI Studio, we first obtained the mean QA value for each streamline and then averaged them for all streamlines between two ROIs. We chose “mean” over “median” because “mean” value is the standard value used for generating connectomes with QA. We had no reason to assume the median would work better. We clarified the description of how QA between ROIs is calculated to construct structural connectomes in the Methods section:

Page #22: “After reconstructing images with GQI, the whole-brain fiber tracking was conducted with DSI-studio (<http://dsi-studio.labsolver.org/>) with quantitative anisotropy (QA) as the termination threshold. QA values were computed in each voxel in their native space for every subject. The tracking parameters were set as the angular cutoff of 60 degree, step size of 1.0mm, minimum length of 30 mm, and maximum length of 300 mm. The whole-brain fiber tracking process was performed with the FACT algorithm until 1,000,000 streamlines were reconstructed for each individual. Here, we used a neonatal AAL-aligned brain parcellation with 90 nodes to construct the structural connectome for each infant. T2-weighted images in native DWI space were used to provide information on region segmentation during the construction of connectomes. The structural connectome for each individual was then constructed with a connectivity threshold of 0.001 and the pairwise connectivity strength was calculated as the average QA value of each fiber connecting the two end regions, which results in a 90x90 adjacent matrix for each participant as the brain structural connectome matrix.”

Minor comment #5

-Were subjects with major lesions in WM excluded? Any other exclusion criteria (e.g. motion)?

Response:

Thank you for bringing up this question. All subjects with major lesions within white matter, cortex, cerebellum, and/or basal ganglia were removed before data releasing by dHCP. Exclusion criterion during preprocessing was based on neighboring DWI correction: 34 scans were excluded due to their low NDC values identified by a median-value based outlier detector. For preterm infants, only subjects with both scan at birth and scan at term-equivalent age were included in this study. In the end, we included 594 scans (448 scans from 448 term infants and 146 repeated scans from 73 preterm infants) in this study. We have added the relevant description about data exclusion criteria to Methods, MRI acquisition section:

“In-scanner head motion was estimated by the SHARD outlier ratio, which is the mean outlier weight of all slices detected in slice-to-volume reconstruction. **A quality check was conducted by neighboring DWI correction (NDC)⁵⁴, leading to 34 scans being excluded due to their low NDC values calculated by a median value based outlier detector.** The accuracy of b-table orientation was examined by comparing fiber orientations with those of a population-averaged template⁵⁵.”

Minor comment #6

-How was head motion calculated? Did the authors use SHARD motion estimates? Was there an association between motion estimates and controllability features? Could descriptive statistics of motion be added to Table 1?

Response:

Thanks for pointing this out. We added more detailed descriptions about head motion in the Methods, MRI acquisition section, second paragraph:

“Diffusion MRI was reconstructed at an in-plane resolution of 1.5mm and slice thickness of 1.5mm. Images were denoised and corrected for motion, eddy current, Gibbs ringing, and susceptibility artifact with the diffusion SHARD pipeline. **In-scanner head motion was estimated by the SHARD outlier ratio, which is the mean outlier weight of all slices detected in slice-to-volume reconstruction.**”

	translation	rotation
Average controllability	r=0.13	r=0.12
Modal controllability	r=0.12	r=0.14

We found that the motion estimations for rotation and translation showed weak correlations with controllability (r’s~0.13, see table above). Nevertheless, we regressed motion from controllability when conducting downstream analysis in the study. Descriptive statistics about in-scanner motion were added to Table 1.

REVIEWER #2

This manuscript takes an intriguing approach, of using network control theory to investigate the controllability of networks in the infant brain, building on the dHCP open dataset. This seems very interesting and would be a novel and important contribution.

However, at present, I have a serious concern about the validity of the analyses. My concerns were raised by the results shown in Figure 2B. It is vanishingly rare in MRI that any two independent measures correlate with $|r| > 0.99$. This is particularly the case for infant MRI data, which are typically of lower signal-to-noise than adult data. My concern was that the two measures are not independent in some way and so the correlation was being inflated. To investigate this, I used the author's code, which they helpfully provided. To investigate the possibility that the average and modal controllability measures are not independent, I generated a random connectivity matrix (here for a notional 50 nodes and 100 subjects). I then ran the first part of the analysis and graphed the output (code attached in Appendix A below).

Even for these completely random data, there is a correlation across nodes, when averaged by subject ($r = -0.993$, $p < 0.001$, scatterplots attached). This shows that the modal and average controllability are not independent. I could not verify that this would be true for random data with the same distribution as the true connectivity, as this was not provided. However, a similar strong correlation persists if the data are made more sparse (by raising the uniform distribution random values to the power of six) or are changed in range from 0-1 to 0-100.

It would be helpful for the authors to consider this issue, and to check the evaluation code in Appendix A. I have not investigated the validity of the later results, but it would be helpful for the authors to also review these with similar concern in mind.

Response:

We apologize for the lack of clarity. We agree that average controllability and modal controllability are not independent. In fact, they are like two sides of one coin and together present a comprehensive picture of the infant brain network under the NCT framework. A large (more negative) correlation between average and modal controllability was also observed (using the same averaging across subjects) in the seminal Gu et al [3] work that introduced NCT to neuroimaging data. In Table S2 of their work, they show a $|r| = 0.97$ between average and modal controllability. This is in line with our values.

It is important to note that average and modal controllability are not completely interchangeable for each individual. Averaging across subjects artificially increases the correlation between modal and average controllability; while on the individual level, the correlations between average controllability and modal controllability for each brain region are much lower than that on the group level ($|r|$'s between 0.7-0.98 compared to the group average of $|r| = 0.99$). Also on the individual level when using the random data, correlations between average and modal controllability were all above $|r| > 0.99$ (see Fig. R1).

Additionally, to test if NCT reflects a unique topology of the infant brain, we constructed null models that preserved the degree distribution on both the real brain data. We then repeated the controllability analysis. After permuting connection, brain networks for over 97% infants showed higher average and modal controllability compared to permuted brain networks. This is not observed in the random networks, indicating that controllability captures unique features from brain network topology, which random data does not have. In sum, while average and modal controllability in brain networks are not independent, each is significantly different from null models and they are significantly less correlated with each other than those from null models.

The analyses associating controllability with longitudinal outcomes show effect size well above chance and differences in brain-behavior associations between the two controllability measures.

To help better frame NCT and our results, we have clarified that modal and average controllability are not independent measures in the introduction, results, and discussion. We have also moved the results regarding modal controllability to the supplementary materials. We have also added results on the null models to the supplement in the "Controlling for network measurements" section. We believe this reformatting helps emphasize the dependent nature of modal and average controllability.

Fig. R1: Histogram of correlations between average and modal controllability at the individual level using random data.

REVIEWER #3

Here the authors present an analysis of publicly available connectome data from infants at various gestational ages and analyse them through the now well-characterised controllability framework, focussing on two controllability measures - average and modal controllability. They then provide descriptive statistics of group differences and longitudinal changes, which are interesting. From the current version, however, it is not clear whether these facilitate any novel insights, since no specific hypotheses are given that were a priori setting out to test specific predictions. Furthermore, little is done statistically to account for a whole host of other changes that occur in networks during this developmental time period to identify which (if any) of the described changes are specific to 'controllability' and how much of these are 'just' corollaries to other developmental changes?

Response:

We thank the reviewer for their insightful comments. While we address these points in greater detail below, we would like to highlight two important changes to the manuscript based on these comments. First, we have modified the introduction to better reflect which questions we aimed to answer in our work. Second, we have included network strength and network density as covariates in all analyses as well as compared controllability and control energy against null models, which preserve the strength and degree distribution of the real data. These additional analyses have strengthened our results and addressed the primary concerns raised by the reviewers.

Major comment #1

- The controllability measured discussed in this paper are quantitative network features, similar to others previously discussed in the infant brain. The author's main hypothesis was that 'controllability developed rapidly across the perinatal period and infants born preterm will have altered controllability' - hypothesis is almost certainly going to be true, given what we already know about changes in network topology during development. It is unclear which aspect of the hypothesis testing specifically leveraged controllability vs just applied controllability as an interesting novel metric from a dataset that was on hand.

Response:

We apologize for the misleading language. We meant this work to be more of an exploratory rather than a confirmatory study. While previous studies have explored a wide range of network features during the perinatal period and in relation to the preterm brains, little is known about how controllability develops over the perinatal period and if these changes are independent of the development of other network features. Confirmatory research is very valuable, as is exploratory research which can provide novel insights that would otherwise remain undiscovered (see for example Tukey, "We Need Both Exploratory and Confirmatory", 1980). We have refocused the last paragraph of the introduction to better reflect the questions that we aimed to understand in our work, as follows:

"We leveraged NCT to address three questions about how the structural foundation facilitates brain dynamics in the infant's brain: (1) whether the infant's brain is controllable and how its

capability in controlling state transitions develops during the early stages of infancy; (2) how much control energy is necessary for transitions to different brain functional network activation states and how preterm birth affects the controllability and control energy cost; (3) how NCT explains the distinct cognitive performance at 18 months old. To answer these questions, we investigated the controllability of structural connectomes during infancy using a large sample of 521 infants, including longitudinal data from 73 preterm infants scanned twice across the perinatal period. We characterized the spatial distribution of regional controllability and how controllability matures from 28 weeks gestation through the first month of postnatal life. We calculated the control energy necessary to drive state transitions to activate different functional brain networks for each individual. Additionally, we investigated the effect of preterm birth on controllability and control energy. Finally, we explored how controllability relates to individual differences in neurodevelopment at 18 months. ”

In addition, the main motivation for using network control theory is that it can be directly associated with changes in brain states. We have included additional analyses to calculate the control energy required to activate specific functional networks (i.e., change brain state; see Reviewer #1’s second major comment). These results further explained how a structural connectome facilitates the switching between states. We believe that this analysis and supporting text better highlight how the questions we are asking require network control theory to answer. Other measures of network topology can not be used to directly simulate changes in brain states.

Major comment #2

- Related to this, the paper as currently presented does not fully test how specific any of its findings are to controllability, and how many may be shaped by e.g. changes in weighted degree or other network measures. This is particularly relevant to aspects where additional inference is drawn - e.g. Fig. 5 - the text here implies that the authors wanted to test 'how regional controllability affects its maturation rate'. The statistics presented don't demonstrate this conclusively, as the reader is not given the statistical tools to see whether the observed could be explained away by e.g. weighted degree as network topological feature that might affect both controllability and regional maturation.

Similarly statistical inference across regions should account for null models to evaluate which changes are 'specific' to controllability, and which ones are corollaries of wider network changes that would occur in any network with the observed changes in e.g. weighted degree. (e.g. see here: <https://www.ncbi.nlm.nih.gov/pmc/articles/PMC7734595/> for examples of null models that might be considered)

Response:

We agree with the reviewer on the importance of showing that any controllability results are specific to controllability and not simply the result of other network changes. We have taken several steps to investigate the specificity of our results. 1) As suggested by Reviewer #1, we have included network density as a covariate in our analyses. As shown in Table S3 and S4, controlling for network strength or network density does not meaningfully change our results and conclusions. 2) We have included network density and network strength as covariates in the additional analyses comparing term to preterms (see Fig. S5 and Fig. S6). Analyses detailed here are presented in the supplementary materials, “Controlling for network measurements”.

Table S3: Controllability maturation during perinatal period controlling for network strength/network density (r-value between controllability and postmenstrual age)

Average Controllability		Modal Controllability
Controlling for sex, brain volume and head motion (shown in the main text)		
Preterm at birth	0.5***	-0.1***
Preterm at TEA	0.11	-0.35**
Term	0.03	-0.50***
Additionally controlling for network strength		
Preterm at birth	0.59***	-0.1,
Preterm at TEA	0.37**	-0.35**
Term	0.5***	-0.53***
Additionally controlling for network density		
Preterm at birth	0.09	-0.23
Preterm at TEA	0.20*	-0.16
Term	0.21***	-0.21***

Table S4: Correlation between regional controllability value and its maturation controlling for network strength/network density (r-value)

Average Controllability		Modal Controllability
Controlling for sex, brain volume and head motion (shown in the main text)		
Preterm at TEA	0.75***	0.5***
Term	0.29**	0.3**
z-stats	5.2***	1.12
Additionally controlling for Network Strength		
Preterm at TEA	0.68***	0.1***
Term	0.2,	0.33**
z-stats	.50***	0.68
Additionally controlling for Network Density		

Preterm at TEA	0.7 ^{***}	0.52 ^{***}
Term	0.25 [*]	0.37 ^{***}
z-stats	5 ^{***}	1.6

Overall, our results suggest that neither network strength nor density can fully account for the observed developmental trends in controllability in the structural brain networks during early infancy. We have included a new section in the results, labeled “Sensitivity analyses” as well as three new supplemental tables and two new supplemental figures.

Additionally, as suggested by the reviewer, to test if NCT reflects a unique topology of the infant brain, we constructed null models that preserved the degree distribution on real brain data. We then repeated the controllability analysis. After permuting connection, brain networks for over 97% infants showed higher average and modal controllability compared to permuted brain networks. These results have also been added to the supplement in the “Controlling for network measurements” section.

Finally, in the new “Control energy cost to activate canonical functional networks” section, we show that the control energy expenditure of the transition from the baseline state to the targeted state was significantly lower in real brain networks than in null model networks that preserved both the strength and degree distribution (Table S7).

Minor comment #1

- developmental description - many aspects of brain anatomy change over the first few decades of life; it is therefore not clear what the authors mean by aspects 'maturing by 40 weeks of gestation' (p1)

Response:

We agree with the reviewer that the sentence was not clear and have removed it for clarity.

Minor comment #2

- wording: control energy is 'biologically evidenced by gray matter integrity and glucose metabolism' - unclear wording

Response:

We have updated this sentence as follows: “Control energy is associated with gray matter integrity, glucose metabolism¹⁴, and efficiency in cognitive execution^{15,16}.”

Minor comment #3

- Figure 2 - not clear to the reader what the difference between A and B is - one is 'whole-brain' (but really just regions averaged by infant); the other one is 'regional' but contains approximately as many dots as plot A? Should there not be more if each region is plotted individually? Are these averaged within region but across infant? (I understand from the methods that this is in fact what has been done, but not clear from the figure alone)

Response:

We apologize for the lack of clarity. For Panel A, the whole brain refers to the average across all regions for a single infant. There is one dot per scan (448 term plus 78 preterm at birth plus 78 preterm at TEA = 604 dots). For Panel B, these values are controllability values for each node averaged across infants and displayed for each group independent (90 term plus 90 preterm at birth plus 90 preterm at TEA = 270 dots). Based on Reviewer's #2 comments, we have removed Panel B and moved the associated analyses to supplemental text. We have also clarified the dot in Panel A in the figure caption.

Minor comment #4

- the inference on infant behaviour is somewhat tricky to relate to the controllability data, and perhaps suggestions about how these could be leveraged into specific testable hypotheses would be helpful

Response:

We agree. Inferences from the association between controllability and cognition are interesting even if interpretation is a bit tricky. These demonstrate how much early brain development predicts later behavior. At the same time, as there is so much brain development between 1-18 months, how does that later development contribute to behavior? To answer these hypotheses, cross-sectional, or even better longitudinal, data through toddlerhood is needed. Once growth curves of both brain and behavior are known, one could investigate which factors (e.g., sleep, maternal depression) might push and pull these curves to better cognitive outcomes. We have added the following to the limitations section: "As such, how controllability develops through later infancy and toddlerhood remains unknown and how these changes support cognitive development. Follow-up longitudinal studies will be needed to clarify how and when controllability reaches similar patterns to those observed in school-age children and how these relate to cognitive outcomes. Once these trajectories are known, future work could begin to investigate how modified environment factors can lead to changes in these trajectories."

We would also like to note that we updated the analysis of BSID-III scores (infant behavior) at 18 months old to be more thorough. All three domains, i.e. cognition, language and motor were taken into consideration when establishing associations between controllability and behavioral scores in a later stage, as shown in the supplementary materials.

Reference

1. Braun, U. et al.: Brain network dynamics during working memory are modulated by dopamine and diminished in schizophrenia. Nat. Commun. 12, 1, 3478 (2021). <https://doi.org/10.1038/s41467-021-23694-9>.
2. Cui, Z. et al.: Optimization of energy state transition trajectory supports the development of executive function during youth. eLife. 9, e53060 (2020). <https://doi.org/10.7554/eLife.53060>.
3. Gu, S. et al.: Controllability of structural brain networks. Nat. Commun. 6, 1, 8414 (2015). <https://doi.org/10.1038/ncomms9414>.

Reviewer #1 (Remarks to the Author):

I commend the authors for their revision of the manuscript, addressing most of my initial comments. I think the energy cost simulations presented in Figure 5 are very interesting and make a nice addition to the paper. I also appreciate the additional analyses exploring the specificity of NCT metrics compared with other measures of network topology.

I only have a few minor additional suggestions:

-I think perhaps authors may want to clarify goal 2 in their revised introduction. As far as I understand they are considering "activation states" of RSNs described as a subset of the anatomical parcellations used. However, goal 2 may imply that they are assessing state transitions using functional connectivity data of participants. Similarly, please consider rephrasing the sentence "We calculated the control energy necessary to drive state transitions to activate different functional brain networks for each individual."

-Please specify in methods section how did you define functional RSNs (visual, somatomotor, dorsal attention, ventral attention, limbic, frontoparietal, default mode network) based on AAL atlas parcellation. I would suggest adding a supplementary table explicitly stating what regions were included in each RSN.

-I think the table showing correlations between network strength/density and AC/MC (response to "major comment #2") is interesting and would be a useful resource for researchers studying early development of controllability in the future. I would suggest incorporating this table in their manuscript, perhaps as a supplementary table.

-I am unsure whether the authors can refer to changes related to "maturation" when referring to cross-sectional differences/correlations. Please consider rephrasing.

-Additional analyses associating controllability with language and motor BSID-III scores aren't mentioned in the body of the manuscript. Perhaps this should be mentioned in "Controllability at birth is associated with cognitive assessments at later ages" section.

-Please consider incorporating in your GitHub repository the code to replicate the new control energy cost experiments performed.

Reviewer #2 (Remarks to the Author):

NCOMMS-23-04790A

Major comments

(1) I thank the authors for the substantial revision, which down-weights the importance of correlations between the average and modal controllability measures, due to their statistical dependence. It does seem appropriate to leave in the comparison across groups of the relationship between these measures (Fig. 2). I have a residual concern which is that the preterm data might have a weaker signal than the term babies, which might be responsible for the changing relationship (note that random noise gives a high negative correlation between average and modal controllability). Preterm babies might give less diffusion signal because their brains are smaller or because of their different brain chemistry (e.g., higher water content, less myelination). The authors may have potentially addressed this, as the control analysis in Table S2 suggests brain size does not substantively affect the distributions. However, other nuisance factors are not controlled for. It would be helpful to calculate metric(s) of diffusion data quality that would change if there was more noise, and repeat the correlation shown in Fig.2 with such covariates. I do not know if network density is such a measure. It would also be helpful to mention in the limitations the issue of signal-to-noise ratio when examining differences between preterms and terms.

(2) Is the relationship between controllability and outcome (BSID III) related to or independent of prematurity? It would be helpful to repeat these correlations with gestational age as a nuisance covariate.

(3) Figure S7. I think the axes have been switched – that the x-axes are the BSID subscales and

the y-axes are controllabilities. This should be checked and the legend and interpretation modified appropriately.

(4) It would be valuable to share the code and instructions on data download and file system organisation such that the results could be readily reproduced.

Minor comments

In the introduction, it would be helpful to give a little more detail in what the average and modal controllability measure. There is a large gap between the high-level one-sentence descriptions in the introduction and the equations in the methods. This might make more interpretable sentences such as on p9 "to support the establishment of a fundamentally efficient brain network"

p9 "t=-3.328" -> given implied sign of comparison in text, remove minus

p17 "white matter architect" -> "white matter architecture"

REVIEWERS' COMMENTS

We thank the reviewers for their helpful comments and suggestions in relation to our revised manuscript, and we have provided point-by-point responses in the following sections.

Reviewer #1 (Remarks to the Author):

I commend the authors for their revision of the manuscript, addressing most of my initial comments. I think the energy cost simulations presented in Figure 5 are very interesting and make a nice addition to the paper. I also appreciate the additional analyses exploring the specificity of NCT metrics compared with other measures of network topology.

I only have a few minor additional suggestions:

-I think perhaps authors may want to clarify goal 2 in their revised introduction. As far as I understand they are considering "activation states" of RSNs described as a subset of the anatomical parcellations used. However, goal 2 may imply that they are assessing state transitions using functional connectivity data of participants. Similarly, please consider **rephrasing the sentence "We calculated the control energy necessary to drive state transitions to activate different functional brain network for each individual."**

Thank you for this kind suggestion. We rephrased that sentence into the following version to avoid confusion.

"We simulated the activation of different functional brain networks on the basis of the individual structural connectome and calculated the control energy theoretically required in each situation."

-Please specify in methods section how did you define functional RSNs (visual, somatomotor, dorsal attention, ventral attention, limbic, frontoparietal, default mode network) based on AAL atlas parcellation. I would suggest **adding a supplementary table** explicitly stating what regions were included in each RSN.

Thanks for the kind suggestion. We have added Table S8. Neonatal 90 node parcellation's regional names and resting-state networks(RSNs) to supplementary information.

-I think the table showing correlations between network strength/density and AC/MC (response to "major comment #2") is interesting and would be a useful resource for researchers studying early development of controllability in the future. I would suggest **incorporating this table in their manuscript, perhaps as a supplementary table.**

Thanks for the kind suggestion. We have added the table as an extension to Table S2: Structural network properties for each infant group (mean(std)) and their associations with controllability in the supplementary information.

-I am unsure whether the authors can refer to changes related to “**maturation**” when referring to cross-sectional differences/correlations. Please **consider rephrasing**.

We agreed that ‘maturation’ can be confusing in the cross-sectional study and have rephrased it to “development” or “changes” in the manuscript.

-Additional analyses associating controllability with language and motor BSID-III scores aren’t mentioned in the body of the manuscript. Perhaps this should be mentioned in “Controllability at birth is associated with cognitive assessments at later ages” section.

Thanks for the kind suggestion. We added one sentence to specify the association between controllability and language and motor BSID-III scores in the end of Results section: Controllability at birth is associated with cognitive assessments at later ages.

-Please consider incorporating in your GitHub repository the code to replicate the new control energy cost experiments performed.

Thanks for the kind suggestion. A more detailed version of code used in the study has been uploaded to GitHub.

Reviewer #2 (Remarks to the Author):

NCOMMS-23-04790A

Major comments

(1) I thank the authors for the substantial revision, which down-weights the importance of correlations between the average and modal controllability measures, due to their statistical dependence. It does seem appropriate to **leave in the comparison across groups of the relationship between these measures** (Fig. 2). I have a residual concern which is that the preterm data might have a weaker signal than the term babies, which might be responsible for the changing relationship (note that random noise gives a high negative correlation between average and modal controllability). Preterm babies might give less diffusion signal because their brains are smaller or because of their different brain chemistry (e.g., higher water content, less myelination). The authors may have potentially addressed this, as the control analysis in Table S2 suggests brain size does not substantively affect the distributions. However, other nuisance factors are not controlled for. It would be helpful to **calculate metric(s) of diffusion data quality that would change if there was more noise, and repeat the correlation shown in**

Fig.2 with such covariates. I do not know if network density is such a measure. It would also be helpful to **mention in the limitations the issue of signal-to-noise ratio when examining differences between preterms and terms.**

Agreed. We added the scatterplot of regional average controllability vs modal controllability in Fig.S1a to indicate the dependence between average controllability and modal controllability.

Furthermore, we also agreed that noise could be an important factor in analysis related to preterm birth effect. To minimize the effect of image noise to downstream analysis, we conducted several steps to denoise the raw diffusion for motion, eddy current, Gibbs ringing, and susceptibility artifact with the diffusion SHARD pipeline during preprocessing. Since the movement of head during scan is one of the main noise sources, we also included the estimation of in-scanner head motion as a covariance in the main analysis. The following several sentences have been added to discussion to better inform about scan-related bias to readers:

“Finally, several demographic and scanning-related factors, including sex, brain volumes, motion artifact and other signal noise during scan, can bias the estimation of developmental trajectories. While we conduct the age-related analysis with the above factors as covariances to limit the impact of these confounding variables, they may still influence the results and would benefit from further study.”

(2) Is the relationship between controllability and outcome (BSID III) related to or independent of prematurity? It would be helpful to **repeat these correlations with gestational age** as a nuisance covariate.

We agreed that prematurity could be a confound in the analysis of association between controllability and BSID-III. The following tables included the correlation results controlling for gestational ages.

BSID-III	Individual AC	Individual MC	Changes in MC for preterm infants
Cognition	r=0.022, p=0.66	r=-0.16, p=0.0016	r=0.29, p=0.036
Language	r=0.00058, p=0.99	r=-0.13, p=0.0094	r=0.0957, p=0.50
Motor	r=0.033, p=0.50	r=-0.079, p=0.11	r=0.093, p=0.51

Compared with the results shown in the Figure S7, we found that the correlations between individual’s average controllability and BSID-III cognitive assessment were not significant after controlling gestational age, but correlations between individual’s modal controllability and BSID-III cognitive assessment were independent of gestational ages.

(3) Figure S7. I think the **axes have been switched** – that the x-axes are the BSID subscales and the y-axes are controllabilities. This should be checked and the legend and interpretation modified appropriately.

Thanks for pointing that out. Corresponding information has been updated in the revised manuscript.

We switched the x-y axes in the revised version for two reasons: 1) BSID-III scores are more reasonable to be considered as outcome/dependent variable/y in the correlation analysis; and 2) to organize the scatterplots in a more clear way.

(4) It would be valuable to **share the code** and instructions on data download and file system organisation such that the results could be readily reproduced.

Thanks for the kind suggestion. A more detailed version of code used in the study has been uploaded to GitHub.

Minor comments

In the introduction, it would be helpful to give a little more detail in what the average and modal controllability measure. There is a large gap between the high-level one-sentence descriptions in the introduction and the equations in the methods. This might make more interpretable sentences such as on p9 “to support the establishment of a fundamentally efficient brain network”

Thanks for the kind advice. We added more explanation in the Discussion section to further interpret the difference between average and modal controllability:

“The contrasting spatial and developmental patterns of average and modal controllability are consistent across our analyses. These patterns likely reflect the distinct but complementary nature of these measures: brain regions with strong average controllability are mainly located in the default mode system and more active during resting state, while strong regions in modal controllability appear more in cognitive control system including fronto-parietal and cingulo-opercular systems and play important roles in tasks requiring high-level cognitive control or task-switching according to previous observations in the young-adult group.”

p9 “ $t=-3.328$ ” -> given implied sign of comparison in text, remove minus
Sorry for the unclarity, the sign has been removed.

p17 “white matter architect” -> “white matter architecture”

Thank the reviewer for pointing that out. The typo has been fixed.

Reviewer #2

I thank the authors for controlling for SNR and gestational age.

Regarding Fig. S7, although the axis labels were changed, I believe incorrectly not switched the data (i.e., scatterplot points), because:

- (1) for each column, the different rows should have the same range on the x axis (as labelled) but they have very different ranges at present (e.g., 1-15 for the top row, 1-40 for the second row)
- (2) the x values appear to be quantised into whole numbers, which would correspond to the BSID III discrete scoring scale, rather than what it is labelled - controllability, which should not be quantised

Thank you for sharing the code, and the extension to the discussion to clarify the measures.